# Hypermethylation of PRKCZ Regulated by E6 Inhibits Invasion and EMT via Cdc42 in HPV-Related Head and Neck Squamous Cell Carcinoma

**DOI:** 10.3390/cancers14174151

**Published:** 2022-08-27

**Authors:** Hao-Fan Wang, Jian Jiang, Jia-Shun Wu, Mei Zhang, Xin Pang, Li Dai, Ya-Ling Tang, Xin-Hua Liang

**Affiliations:** 1State Key Laboratory of Oral Diseases and National Clinical Research Center for Oral Diseases, Department of Oral and Maxillofacial Surgery, West China Hospital of Stomatology, Sichuan University, Chengdu 610041, China; 2Department of Head and Neck Surgery, Sichuan Cancer Hospital & Institute, Sichuan Cancer Center, School of Medicine, University of Electronic Science and Technology of China, Chengdu 610041, China; 3State Key Laboratory of Oral Diseases and National Clinical Research Center for Oral Diseases, Department of Pathology, West China Hospital of Stomatology, Sichuan University, Chengdu 610041, China

**Keywords:** human papillomavirus (HPV), head and neck squamous cell carcinoma (HNSCC), PRKCZ, DNA methylation

## Abstract

**Simple Summary:**

HPV+ and HPV- HNSCC share distinct epigenetic characteristics and clinicopathological features. The aim of our study was to assess whether DNA methylation plays a role in the progression of HPV+ HNSCC. We used a HumanMethylation450 BeadChip array (Illumina) to identify PRKCZ genes exhibiting different levels of DNA methylation between HPV+ and HPV- HNSCC. PRKCZ acts as a potent tumor promoter in HPV+ HNSCC. These findings may provide a possible explanation for the differences in the clinicopathological characteristics between HPV+ and HPV- HNSCC and promising ideas for the treatment of HPV+ HNSCC.

**Abstract:**

Purpose: To study the role of target genes with aberrant DNA methylation in HPV+ HNSCC. Methods: A HumanMethylation450 BeadChip array (Illumina) was used to identify differentially methylated genes. CCK-8, flow cytometry, wound healing, and cell invasion assays were conducted to analyze the biological roles of PRKCZ. Western blot, qRT-PCR, immunohistochemistry, and animal studies were performed to explore the mechanisms underlying the functions of PRKCZ. Results: We selected PRKCZ, which is associated with HPV infection, as our target gene. PRKCZ was hypermethylated in HPV+ HNSCC patients, and PRKCZ methylation status was negatively related to the pathological grading of HNSCC patients. Silencing PRKCZ inhibited the malignant capacity of HPV+ HNSCC cells. Mechanistically, HPV might promote DNMT1 expression via E6 to increase PRKCZ methylation. Cdc42 was required for the PRKCZ-mediated mechanism of action, contributing to the occurrence of epithelial-mesenchymal transition (EMT) in HPV+ HNSCC cells. In addition, blocking PRKCZ delayed tumor growth in HPV16-E6/E7 transgenic mice. Cdc42 expression was decreased, whereas E-cadherin levels increased. Conclusion: We suggest that PRKCZ hypermethylation induces EMT via Cdc42 to act as a potent tumor promoter in HPV+ HNSCC.

## 1. Introduction

Head and neck squamous cell carcinoma (HNSCC) is the most common malignant tumor stemming from the oral cavity, with approximately 890,000 new cases and 450,000 deaths occurring worldwide in 2018 [1,2]. Despite constant improvements in treatments, the 5-year survival rate of HNSCC patients remains unsatisfactory at only approximately 50% [3,4]. In addition to classic carcinogenic factors such as smoking, drinking, and chewing betel nuts, human papilloma virus (HPV) infection is considered another main carcinogenic factor in HNSCC [5]. Of all the HPV types with the ability to result in HNSCC, HPV 16 is the most frequent, accounting for approximately 80% of cases, followed by HPV 18, which is responsible for approximately 3% of cases [6]. The oncoproteins E5, E6, and E7 play vital nonnegligible roles in HPV16-mediated carcinogenesis [7]. E7 binds to and induces degradation of the retinoblastoma protein (pRb) to dysregulate the G1/S-phase transition, whereas E6 targets the degradation of the p53 tumor suppressor protein to disrupt pro-apoptotic signaling. In addition, E5 drives cell proliferation. Thus, a persistent HPV infection makes cancers much more likely to develop and progress [8]. Compared with HPV- HNSCC, HPV+ HNSCC has been shown to exhibit a better prognosis and sensitivity to radiotherapy and chemotherapy [9], perform a lower mutational burden, expressed a wild TP53 [10,11], have more tumor-infiltrating lymphocytes [12] as well as the less hypoxic [13], however, why HPV+ HNSCC performs better prognosis and the sensitivity of radiotherapy and chemotherapy still remains unclear.

The accumulation of epigenetic alterations, including aberrant DNA methylation, is an important carcinogenic feature of HNSCC [14]. DNA methylation refers to a chemical modification process in which a specific base of a DNA sequence obtains a methyl group by covalent bonding under the catalysis of DNA methyltransferase (DNMT) [15,16]. Abnormal DNA methylation can lead to the transformation of normal cells into tumor cells, resulting in the occurrence and development of malignant tumors. The hypermethylation of tumor suppressor genes, causing transcription inhibition, is a key process of tumorigenesis [17,18]. Global hypomethylation in DNA and local hypermethylation in specific CpG islands are special hallmarks of cancers [19]. Global hypomethylation could rise a significant 1.6-fold for HNSCC [20]. In addition, the hypermethylation of several tumor suppressors such as *CDKN2A*, DAPK, MGMT, and E-cadherin is crucial in HNSCC progression [21].

The *PRKCZ* gene is located on human chromosome 1 (1p36.33–p36.2) and encodes the following two proteins: Protein kinase M zeta (PKM-ζ) and Protein kinase C zeta (PKC-ζ, PRKCZ). PKM-ζ is a constitutively active form of atypical PKC that is expressed only in the nervous system and is involved in long-term memory maintenance [22]. PKC-ζ can be expressed in a variety of tissues and is involved in multiple signal transduction pathways, such as activation of the ERK/MAPK cascade and the transcription factor NF-κB [23], which may contribute to the role of PRKCZ as a fundamental regulator of tumorigenesis [24]. PRKCZ has been considered a significant mediator with a tumor-promoting or tumor-suppressing role in different cancers [25]. However, evidence regarding the role of PRKCZ methylation in cancers remains limited.

In this study, we used the HumanMethylation450 BeadChip array (Illumina) to identify the *PRKCZ* gene with different degrees of DNA methylation between HPV+ and HPV- HNSCC for the first time. Our data further illustrated that blocking PRKCZ expression could inhibit the proliferation, migration, and invasion abilities and increase the apoptosis ratio of HPV+ HNSCC cells. In addition, downregulated PRKCZ expression could damage the process of epithelial-mesenchymal transition (EMT). These findings may provide a possible explanation for the differences in the clinicopathological characteristics between HPV+ and HPV- HNSCC, and promising ideas for the treatment of HPV+ HNSCC. 

## 2. Materials and Methods

Patients and clinical specimens 40 primary HNSCC tissue specimens, including 12 paraffin-embedded tissue specimens and 28 fresh tissue specimens, were derived from patients without history of previous treatments or autoimmune diseases who had accepted radical surgery at West China Hospital of Stomatology, Sichuan University. All patients gave informed consent. These specimens have been pathologically identified as HNSCC and divided into different grades on the basis of the International Union for Cancer Control’s latest standards. The clinicopathological characteristics of these patients are provided in Appendix A. All samples were fixed with 10% formalin, paraffin-embedded, or frozen at −80 °C for RNA and DNA extraction, or for subsequent tests.

### 2.1. HumanMethylation450 BeadChip Array (Illumina)

An HumanMethylation450 BeadChip array (Illumina, Design ID: OE2015Q1029) of clinical specimens from HNSCC patients who had accepted radical surgery at West China Hospital of Stomatology, Sichuan University, including 3 HPV-positive cases and 3 HPV-negative cases, was provided by oeBiotech Limited Company. The clinicopathological characteristics of these 6 patients were provided in Appendix A.

### 2.2. DNA Isolation and Methylation Analysis

Genomic DNA from 12 paraffin-embedded tissue specimens and 28 fresh tissue specimens was isolated using DNA Extraction Kit (Tiangen, Beijing, China). Bisulfite conversion of DNA was carried out using the EZ DNA Methylation Kit (Zymo Research, Orange, CA, USA) following the manufacturer’s procedure. Gene methylation was analyzed by Methylamp MS-qRT-PCT Fast Kit (Epigentek, Farmingdale, NY, USA), which was performed on ABI PRISM 7900 sequence detection system (Thermo Fisher Scientific, Waltham, MA, USA) at 95 °C 7 min → (95 °C 10 s → 55 °C 10 s → 72 °C 8 s) × 40 cycles → 72 °C 1 min. The β-actin (ACTB) was used as an endogenous control. The primers for qMSP are as follows: 

PRKCZ-F: TTTTTTTATAGGGGATTTTGGATTC; 

PRKCZ-R: CACACTTACGAACTAAAACTACGAA;

ACTB-F: TGGTGATGGAGGAGGTTTAGTAAGT;

ACTB-R: AACCAATAAAACCTACTCCTCCCTTAA

All experiments were repeated three times at least and the methylation degree of PRKCZ was analyzed using 2^−(ΔCt PRKCZ–ΔCt ACTB)^ method [26].

### 2.3. qRT-PCR

Total RNA of tissue specimens was extracted according to Trizol instructions. The reverse transcription was carried out using SureScript™ First-Strand cDNA Synthesis Kit (GeneCopoeia, Rockville, MD, USA) was carried out for reverse transcription reaction. Then qRT-PCR was performed with BlazeTaq™ SYBR^®^ Green qPCR mix 2.0 (GeneCopoeia) on ABI PRISM 7900 sequence detection system. Reaction system: 12 μL DEPC water, 4 μL 5 × BlazeTaq qPCR Mix, 1 μL Forward Primer (10 μM), 1 μL Reverse Primer (10 μM), 2 μL template cDNA, final volume was 20 μL. Reaction conditions: 95 °C 30 s → (95 °C 10 s → 60 °C 30 s) × 40 cycles. Specific primers of each gene for qRT-PCR were shown in Appendix A. All experiments were repeated three times at least and the relative expressions of detected genes were calculated via 2^−^^△△CT^ method [27].

### 2.4. Western Blotting

Total proteins of tissue specimens were prepared via RIPA lysis buffer. Next, proteins were separated through sodium dodecyl sulfate–polyacrylamide gel electrophoresis. Then, gel was transferred onto a PVDF membrane (Millipore, Boston, MA, USA). After that, membranes with target protein were blocked in 5% defatted milk and incubated with specific primary antibody overnight at 4 °C. Primary antibodies were listed as follows: anti-PKC-ζ (1:1000, rabbit anti-human, Proteintech, Chicago, IL, USA), anti-Cdc42 (1:1000, rabbit anti-human, Proteintech), anti-E-cadherin (1:5000, mouse anti-human, Cell Signaling Technology Inc., Danvers, MA, USA), anti-N-cadherin (1:1000, rabbit anti-human, Proteintech), anti-Vimentin (1:1000, rabbit anti-human, Proteintech), anti-HPV16 E6 (1:1000, mouse anti-human papillomavirus, Abcam), anti-HPV16 E7 (1:1000, mouse anti-human papillomavirus, Abcam, Cambridge, UK), anti-DNMT1 (1:500, rabbit anti-human, Proteintech), anti-DNMT3a (1:10,000, rabbit anti-human, Proteintech), anti-DNMT3b (1:3000, rabbit anti-human, Proteintech). After incubating goat anti-rabbit or goat anti-mouse secondary antibody (MultiSciences, Hangzhou, China), membranes with target protein incubated with the chemiluminescence reagent (Beyotime Biotechnology, Haimen, China) were visualized via ChemiDoc XRS+ System (Bio-Rad, Hercules, CA, USA). All original Western blots can be found in Appendix A.

### 2.5. Immunohistochemistry (IHC)

Formalin-fixed paraffin-embedded sections were firstly dewaxed in xylene and rehydrated in graded alcohol solutions. Then antigen retrieval was performed by boiling for 3 min in 0.01 M citrate buffer (pH = 6.0), and sections were immersed in 3% hydrogen peroxide for 10 min and in PBS for 5 min twice. After incubating with 5% goat serum for antigen blocking, the sections were incubated with the primary antibody at 4 °C overnight in a moist chamber. Additionally, the following primary antibodies were listed: anti-PKC-ζ (1:300, rabbit anti-human, Proteintech), anti-P16 (1:500, rabbit anti-human, Bioss, Boston, MA, USA), anti-Cdc42 (1:200, rabbit anti-human, Proteintech), anti-E-cadherin (1:500, mouse anti-human, Cell Signaling Technology Inc.), anti-N-cadherin (1:100, rabbit anti-human, Proteintech), anti-Vimentin (1:100, rabbit anti-human, Proteintech). After washing with PBS twice, the sections were incubated with biotinylated anti-mouse/rabbit IgG and sequent with streptavidin-biotin peroxidase both for 15 min. DAB was added to detect the primary antibody, and hematoxylin was used to stain the nucleus. As negative controls, sections were analyzed in parallel except incubating with isotype-specific immunoglobulin (IgG) instead of the primary antibody. Quantification was evaluated by three independent investigators who were blinded to patient characteristics. According to the degree of staining in ten randomly selected fields at 400× magnification, the level of target protein was assigned as 0 (negative), 1 (weak), 2 (moderate), and 3 (strong).

### 2.6. Cell Culture and Reagents

HPV16-positive human HNSCC cell lines SCC47 and SCC090, HPV-negative human HNSCC cell lines Cal27, SCC25, and SCC9 were obtained from State Key Laboratory of Oral Disease, Sichuan University. SCC47, SCC090, Cal27, and SCC9 were cultured in DMEM (Gibco, Grand Island, NY, USA) containing 10% fetal bovine serum (FBS, Gibco). SCC25 was cultured in DMEM-F12 (Gibco) containing 10% fetal bovine serum (FBS, Gibco). All of them were maintained at 37 °C in a humidified atmosphere with 5% CO_2_.

### 2.7. Transfections

GenePharma Co., Ltd., Suzhou, China provided the design and composition services of small-interfering RNAs (siRNAs) targeting E6, E7, and PRKCZ. EndoFectin^TM^ (GeneCopoeia) was used for transfections according to the manufacturer’s instructions. Further experiments were conducted 48 h after transfection. The plasmids for HPV16 E6 and E7 overexpression and negative control plasmid were designed by GenePharma Co., Ltd., China. Stably transfected Cal27 with E6 and E7 overexpression were derived from the parental cells by a fluorescence microscope (Olympus BX51, Olympus Corporation, Shinjuku City, Tokyo, Japan) detection and puromycin (Sigma-Aldrich, St. Louis, MO, USA) selection. The sequences of siRNAs are listed in Appendix A. 

### 2.8. Cell Proliferation Assay and Flow Cytometry-Based Apoptosis Analysis

CCK-8 assay was used to test cell proliferation capacity. Treated cells were seeded in 96-well plates at a density of 1 × 10^3^ (cells/well) and cultured at 37 °C in a humidified atmosphere with 5% CO_2_ for 1–5 days. Subsequently, cells were treated with 10 μL CCK-8 reaction solution (Hanbio Inc., Shanghai, China) and incubated for 1 h in cell constant temperature incubator. Then, the light absorption value (OD value) at 450 nm was measured. Each assay was carried out in triplicate.

Flow cytometry using the Annexin V-FITC Apoptosis Detection Kit (Invitrogen) was for the test of cell apoptosis. Treated cells were seeded in 6-well plates and digested after 48 h. Then, cells were resuspended with 1X Binding Buffer. Stained with 5 μL of Annexin V-FITC Conjugate and 5 μL of Propidium Iodide Solution, cells were incubated for 30 min at room temperature under dark conditions. After washing and resuspending, the percentage of stained cells was analyzed using a FACScalibur (Becton-Dickinson). Experiments were conducted in triplicate.

### 2.9. Wound Healing Assay and Cell Invasion Assay

Appropriate cells were seeded in 6-well plates to obtain 100% confluence next day. After 24 h of starvation in serum-free medium, a pipette tip was used to scratch the cells and form the clearance space with suitable width. After washing with PSC to clear out the dead cells, photomicrographs of clearance space were obtained at the timepoint of 0 and 48 h. Closed scratch areas were calculated by ImageJ software. Each experiment was conducted in triplicate.

5 × 10^4^ cells were seeded into the upper chamber of 24-well transwell plates (pore size 8 μm, Millipore) with Matrigel-coated membrane (BD Bioscience). After incubation for 24 h in cell constant temperature incubator, the medium in upper chamber was changed into serum-free, while that in lower chamber contained 10% FBS. Then, after 24 h incubation, cells that invaded through the Matrigel-coated membrane were fixed in 4% paraformaldehyde for 30 min and stained with 0.1% crystal violet and counted under a microscope. Five randomly selected fields were used to count the number of stained cells. Assays were conducted in triplicate.

### 2.10. Animal Experiments

For 4-nitroquinoline 1-oxide (4-NQO)-indued HNSCC model of transgenic mice, a total of fifteen 7-week-old HPV16-E6/E7 transgenic mice after one week of acclimation (Cyagen, ID: TOS150814BA1) were used and divided randomly into the following three groups: 4-NQO + PKC-ζ pseudosubstrate inhibitor group (*n* = 5), 4-NQO + PRKCZ siRNA group (*n* = 5) and 4-NQO group (*n*= 5). Mice were fed with distilled drinking water containing a concentration of 100 μg/mL 4-NQO (Sigma-Aldrich) solution for 10 weeks, then distilled water for another 10 weeks to generate HNSCC. After that, PKC-ζ pseudosubstrate inhibitor (Cayman) was injected near the tumor every three days at 100 mg/kg in 4-NQO + PKC-ζ pseudosubstrate inhibitor group. The 4-NQO + PRKCZ siRNA group was injected with PRKCZ siRNA according to the same concentration and method, while 4-NQO group was injected with DEPC water. Mice were killed by cervical dislocation after 3 weeks, and tongues were collected. Specimens bisected longitudinally were fixed with 10% formalin, paraffin-embedded, or frozen at −80℃. The sequence of PRKCZ siRNA was GCCATGAAGGTGGTAAAGAAG.

### 2.11. Statistical Analysis

Means comparisons were performed with Student *t*-tests or one-way ANOVA. The association between PRKCZ methylation and mRNA expression was assessed via 2-tailed Pearson’s statistics. All cellular experiments were conducted independently at least three times and in triplicate each time. GraphPad Prism 7.0 (GraphPad Software) was used for processing all data, and values were presented as means ± SD. *p* < 0.05 was considered to be statistically significant.

## 3. Result

### 3.1. PRKCZ Was Hypermethylated in HPV+ HNSCC Compared with HPV- HNSCC

To identify the differentially methylated genes between HPV+ and HPV- HNSCC, a HumanMethylation450K BeadChip array (Illumina) was performed. According to the analyzed data, 1344 genes were hypermethylated and 754 genes were hypomethylated in HPV+ HNSCC, compared with HPV- HNSCC (Figure 1A). We collected 25 significantly hypermethylated and 25 obviously hypomethylated genes in HPV+ HNSCC based on array data (Figure 1B, Appendix A). Furthermore, we performed KEGG analysis on these selected genes. PKRCZ was found to be the only gene associated with HPV infection (Figure 1C and Appendix A). Therefore, PRKCZ was selected as the focus of our study.

We used results from the TCGA database, paraffin-embedded tissues, and fresh tissues to verify the accuracy of the data obtained from the HumanMethylation450 BeadChip array. According to results from the TCGA database [28,29], PRKCZ mRNA expression was higher in HPV+ HNSCC compared with HPV- cases and the degree of PRKCZ methylation was negatively associated with its mRNA level (Figure 1D). However, no significant difference in the methylation degree of PRKCZ was noted between HPV+ HNSCC and HPV- cases. In addition, we found that PRKCZ was hypermethylated in HPV+ HNSCC paraffin-embedded tissues and fresh tissues, compared with HPV- samples via qMSP (Figure 1E). These findings were consistent with the results from the array analysis. In addition, PRKCZ mRNA expression in HPV+ HNSCC fresh tissues was significantly lower than that in HPV- tissues (Figure 1F). A negative correlation between PRKCZ methylation degree and the mRNA expression level was also observed in fresh tissues (r = −0.3788, *p* = 0.0469) (Figure 1G). This difference in the level of PRKCZ methylation in the TCGA database and our data may be due to the use of different detection methods to assess PRKCZ methylation degree and HPV status. Furthermore, the rate of positive PRKCZ staining, which was mainly localized in the cytoplasm, was higher in HPV+ HNSCC. In HPV+ HNSCC, weak, moderate, and strong positive rates were 61.5% (8/13), 15.4% (2/13), and 23.1% (3/13), respectively. In HPV- HNSCC, weak, moderate and strong positive rates were 17.1% (6/35), 40.0% (14/35) and 42.9% (15/35), respectively (Figure 1H,I).

Furthermore, we detected the differences in PRKCZ methylation and mRNA levels between HPV+ HNSCC cells (SCC47 and SCC090) and HPV- HNSCC cells (Cal27, SCC25 and SCC9). Consistent with the results above, HPV+ HNSCC cells had increased methylation degrees and decreased mRNA and protein expressions (Figure 2A,B). To further verify the correlation between PRKCZ methylation degree and mRNA, and protein expression, we treated cells with 5-aza-2’-deoxycytidine (5-aza), a potent DNA methylation-specific inhibitor [30,31]. As the concentration of 5-aza increased, the degree of PRKCZ methylation decreased, whereas PRKCZ mRNA and PKC-ζ protein levels increased in SCC47 (Figure 2C,D) and Cal27 (Figure 2E,F) cells. Taken together, we found that PRKCZ was hypermethylated in HPV+ HNSCC, which was related to decreased mRNA and protein levels.

### 3.2. Relationship between PRKCZ Methylation and Clinicopathological Parameters of HNSCC Patients

We tested the predictive value of PRKCZ methylation status for relevant clinical and pathological parameters of collected HNSCC specimens, including 12 paraffin-embedded samples and 28 fresh samples. The PRKCZ methylation levels of each specimen were determined via qMSP. The associations between PRKCZ methylation status and patients’ sex, age, T stage, lymph node metastasis, and tumor recurrence were not significant (*p* = 0.3138, *p* = 0.4679, *p* = 0.2365, *p* = 0.5322, and *p* = 0.5827, respectively), whereas PRKCZ methylation status was significantly related to the pathological grading of HNSCC patients (*p* = 0.0386). Patients with PRKCZ hypermethylation seemed to have worse pathological manifestations (Figure 2G). The fresh samples we collected were obtained within 3 years; thus, we did not assess the prognostic value of PRKCZ methylation status. Thus, we used bioinformatics to assess the prognostic value of PRKCZ methylation and mRNA from the TCGA and GEO databases. These databases showed that low PRKCZ expression in HPV+ HNSCC patients seems to have a better prognosis; however, HPV+ HNSCC patients with PRKCZ hypermethylation and low expression have much better survival. In addition, for HPV- HNCSS patients, PRKCZ expression and methylation level have no significant effects on the prognosis (Figure 2H,I). The relations between PRKCZ methylation status and relevant clinical and pathological parameters of HPV+ and HPV- patients were presented in Appendix A. No differences between PRKCZ methylation status and relevant clinical and pathological parameters were noted in HPV+ patients and HPV- patients.

### 3.3. Blocking PRKCZ Inhibited HPV+ HNSCC Cells Proliferation, Invasion, and Migration, Promoted Apoptosis

To provide more powerful proof of the HPV effect on PRKCZ methylation and the PRKCZ effect on HPV+ HNSCC, we established a new HPV+ HNSCC-like cell line termed Cal+E6E7 cell by transfecting Cal27 cells with E6 and E7 lentivirus plasmids. The Western blot results revealed E6 and E7 expression in the new cell line, illustrating successful construction (Figure 3A). Next, two pairs of siRNAs targeting different sites of PRKCZ mRNA were used to silence expression. Compared with the control siRNA, siRNA1 and siRNA2 reduced PRKCZ mRNA relative expression levels in SCC47 and Cal27+E6E7 cells by 25% and 38%, and 32% and 45%, respectively. Western blot experiments also demonstrated the silencing effects of both pairs of siRNAs on PKC-ζ protein expression (Figure 3B). Thus, siRNA1 was selected for follow-up experiments.

After transfecting two types of HPV+ HNSCC cells (SCC47 and Cal27+E6E7) with siRNA1, we found that the OD values on Days 3, 4, and 5 were significantly lower in the transfected cells compared with two cells transfected with control siRNA. This finding suggests that silencing PRKCZ impairs the proliferation abilities of HPV+ HNSCC cells (Figure 3C). We applied wound healing (Figure 3D and Appendix A) and transwell invasion (Figure 3E and Appendix A) assays to investigate the effect of PRKCZ on the migration and invasion of SCC47 and Cal27+E6E7 cells. The results showed that the wound healing areas of HPV+ HNSCC cells transfected with siRNA1 and control siRNA were 15.150% and 23.216% (SCC47), as well as 17.801% and 25.050% (Cal27+E6E7), respectively. The numbers of invaded cells were 498 and 756 (SCC47), as well as 287 and 473 (Cal27+E6E7), respectively. These data suggest that decreased PRKCZ levels dramatically reduce the migratory and invasive capacities of SCC47 and Cal27+E6E7 cells. Regarding the role of PRKCZ in cell apoptosis, flow cytometry analysis showed that after transfection with siRNA1 for 48 h, the apoptosis rates of SCC47 and Cal27+E6E7 cells were 27.10% and 22.40%, respectively, whereas those of these two cells transfected with control siRNA were 12.04% and 13.21%, respectively (Figure 3F).

In addition, we used a specific inhibitor of PRKCZ, a PKC-ζ pseudosubstrate inhibitor, which decreased PRKCZ mRNA level in SCC47 cells to 10% (Figure 3G). Similar to the results above, the OD values on Days 3, 4, and 5 of SCC47 cells treated with the pseudosubstrate inhibitor were significantly reduced compared with those of normal SCC47 (Figure 3H). In addition, the percentage of wound healing areas (Figure 3I and Appendix A) and the number of invading SCC47 cells (Figure 3J and Appendix A) treated with the pseudosubstrate inhibitor were 14.571% and 271, respectively, while those of normal SCC47 cells were 24.174% and 478, respectively. These results suggest that the pseudosubstrate inhibitor reduces the proliferation, invasion, and migration behaviors of SCC47 cells. In addition, after treatment with the pseudosubstrate inhibitor for 48 h, the apoptosis rate of SCC47 cells was increased (24.30%), compared with that of control cells (10.77%) (Figure 3K). All the results suggested that PRKCZ could promote HPV+ HNSCC cell proliferation, invasion, and migration potential and inhibit apoptosis.

### 3.4. Methylation Degree of PRKCZ Was Increased by Partly DNMT1 via HPV E6

E6 and E7, two main well-known carcinogens of HPV, target host tumor suppressor genes or proteins to drive carcinogenesis and progression [32]. Thus, we investigated whether HPV affected the methylation degree of PRKCZ via E6 or E7. We compared the methylation degree of PRKCZ between Cal27 and Cal27+E6E7 cells. It was suggested that after transfection with E6 and E7 lentivirus plasmids, PRKCZ methylation levels were increased, whereas PRKCZ mRNA levels decreased (Figure 4A). To further determine whether E6 or E7 was more powerful, E6 and E7 levels in SCC47 were inhibited using respective siRNAs. Additionally, the most effective siRNAs for E6 and E7 were selected by qPCR and WB experiments (Figure 4B). We found that PRKCZ methylation levels decreased more obviously when E6 was inhibited. Accordingly, the mRNA level increased even more (Figure 4C).

The DNMT family has been demonstrated to act as irreplaceable and crucial enzymes involved in DNA methylation [33,34]. Thus, we hypothesized that HPV mediates PKRCZ methylation by regulating the expression of the DNMT family. The expression levels of several main members of the DNMT family (DNMT1, DNMT3a, DNMT3b) were detected after E6 or E7 was inhibited in SCC47 cells. The results showed that when E6 was inhibited, DNMT1 levels decreased, whereas changes in the levels of DNMT3a and DNMT3b were not obvious. The same phenomenon was observed when E7 was inhibited. In addition, the role of E6 on DNMT1 was more potent than that of E7 (Figure 4D,E). Overall, it was demonstrated that in SCC47 cells, HPV at least partly regulated DNMT1 expression through E6, leading to an increase in PRKCZ methylation and a decrease in PRKCZ mRNA expression.

### 3.5. PRKCZ Regulated Rap Signaling Pathway to Mediate EMT of HPV+ HNSCC Cells

To investigate the internal mechanisms by which PRKCZ influences HPV+ HNSCC cell biological behaviors, we performed a KEGG enrichment analysis of PRKCZ. The results suggested that among the pathways in which PRKCZ is involved, the Rap signaling pathway was ranked 7th among the top 20 KEGG pathways for abnormal gene methylation enrichment (Figure 5A and Appendix A). In addition, according to the results described above, we found that HPV+ HNSCC cell migration and invasion were significantly inhibited after silencing PRKCZ expression. It is well known that EMT is a vital process in tumor invasion and metastasis [35,36]. Thus, we hypothesized that PRKCZ regulated the Rap signaling pathway to mediate EMT, ultimately influencing HPV+ HNSCC cell migration and invasion.

First, we detected the expression changes of several key genes involved in the Rap signaling pathway (Epac1, Rac1, Rap1A, Rap1B, Cdc42, RhoA, and RhoB) in HPV+ HNSCC cells after PRKCZ expression was blocked. The qPCR results suggested that after PRKCZ silencing, Epac1, Rac1, Rap1A, Rap1B, Cdc42, and RhoB mRNA expression decreased, whereas the change in RhoA was not significantly different. In addition, Cdc42 mRNA levels changed most dramatically (Figure 5B). Cdc42 protein levels were also inhibited (Figure 5C). Next, the expression levels of several key genes involved in EMT were detected. After inhibiting PRKCZ, E-cadherin mRNA and protein expression levels increased, whereas N-cadherin and Vimentin mRNA and protein expression levels decreased (Figure 5D). This result suggested that silencing PRKCZ could inhibit the occurrence of EMT in HPV+ HNSCC cells, thus inhibiting cell migration and invasion.

### 3.6. Blocking PRKCZ Delayed the Tumor Growth of HPV16-E6/E7 Transgenic Mice

The 4-NQO supplied in drinking water is widely used as a carcinogen to induce cancers of the oral cavity (HPV-) in rodents due to the similar histological and molecular changes observed in human oral carcinogenesis [37]. In our experiment, we used 4NQO-induced HPV16-E6/E7 transgenic mice tongue tumor model, which was obtained from Rosa26-E6-E7 constitutive knock-in homozygous C57BL/6 mice. The subsequent breeding and identification of transgenic mice were completed by our research group in the early stages [38,39]. After drinking water with 4-NQO for 10 weeks followed by distilled water for an additional 10 weeks, PKC-ζ pseudosubstrate inhibitor and PRKCZ siRNA were injected near the tumor at 3-day intervals, and the mice were assessed after 3 weeks (Figure 6A). The control group was treated with DEPC water. HE staining results demonstrated the successful establishment of tongue squamous carcinoma via 4-NQO (Figure 6B). Tumor lesions in the tongue were more obvious in the DEPC-treated group compared with the inhibitor-treated and siRNA-treated groups. In addition, tongue lesions were slightly more apparent in the inhibitor-treated group compared with the siRNA-treated group, but the difference was not statistically significant. The tumor size was (1.852 ± 0.43) mm^3^ on average in inhibitor-treated mice and (1.799 ± 0.630) mm^3^ in siRNA-treated mice, whereas that in DEPC-treated mice was (5.108 ± 1.962) mm^3^ (Figure 6C). These results revealed that blocking PRKCZ could weaken the formation of tongue lesions induced by 4-NQO in HPV16-E6/E7 transgenic mice.

Then, IHC staining demonstrated that Cdc42 expression decreased in tumors from the inhibitor-treated and siRNA-treated groups, whereas E-cadherin levels increased (Figure 6D). These results indicated that suppressing PRKCZ could delay 4NQO-induced tongue tumor growth by promoting Cdc42 expression and down-regulating E-cadherin expression.

## 4. Discussion

The importance of DNA methylation in host–viral interactions and cancer development is self-evident [40,41]. The methylation landscapes between HPV+ and HPV- HNSCC are quite different. In this study, we identified PRKCZ as a differentially methylated gene between HPV+ and HPV- HNSCC using a HumanMethylation450K BeadChip array (Illumina). We addressed, for the first time, the existence of an HPV-mediated PRKCZ hypermethylation signature in HPV+ HNSCC, which was largely due to the role of E6 in stimulating DNMT1 expression. Moreover, we suggest that PRKCZ plays a tumor-promoting role in HPV+ HNSCC via Cdc42, inhibiting E-cad expression, promoting N-cad and Vim expression, and finally contributing to EMT.

HPV has a specific effect on shaping the DNA methylome of HNSCC [42]. In addition, the accumulation of abnormal epigenetic alterations may result, at least partly, in the differences between HPV+ HNSCC and HPV- HNSCC [43]. HPV+ HNSCC and HPV- HNSCC show different DNA methylation profiles. HPV- HNSCC is primarily characterized by genome-wide hypomethylation, whereas HPV+ counterparts seem to have increased maintenance of global methylation and exhibit a greater connection with promoter hypermethylation [44]. In our study, we found that compared with HPV- HNSCC, HPV+ HNSCC exhibited PRKCZ hypermethylation, which was associated with decreased PRKCZ mRNA and PKC-ζ protein expression. Additionally, decreased PRKCZ expression was responsible for the impaired HPV+ HNSCC cell proliferation, migration, and invasion. Thus, we suggest that the impaired malignant behaviors of HPV+ HNSCC cells due to PRKCZ hypermethylation might explain why HPV+ HNSCC has a better prognosis. In addition, PRKCZ hypermethylation was associated with the pathological grade of HNSCC patients, but not with sex, age, T stage, lymph node metastasis, or tumor recurrence. The results from cell experiments and clinical specimens were slightly contradictory. There are two possible explanations. First, the insufficient number of clinical specimens in our study might explain the difference. The number of patients enrolled in our study was limited (40 patients), and the follow-up date was short (the longest was 78 months, the shortest was 13 months, and the average was 35.6 months). Second, the tumor microenvironment is very complex. In vitro cell experiments cannot accurately reflect changes in the body. Whether the effect of PRKCZ on some important tumor stromal cells, such as cancer-associated fibroblasts, can affect or even counteract its effect on tumor cells, which needs further research.

PRKCZ was the only gene associated with HPV infection among 50 selected genes with the most obvious methylation differences between HPV+ HNSCC and HPV- HNSCC according to array data. This finding was further supported by data obtained from TCGA, clinical specimens, and cell lines, revealing that PRKCZ was hypermethylated in HPV+ HNSCC. However, the difference in the level of methylation observed in TCGA data was not significant, which may be due to a discrepancy in the PRKCZ methylation degree and HPV status. PRKCZ hypermethylation plays a vital role in several diseases, such as type 2 diabetes mellitus [45], CTCF deletion syndrome [46], and postmenopausal osteoporosis [47]. However, to the best of our knowledge, only one study has reported the role of PRKCZ in cancers. In COX-2-induced hepatocellular carcinoma, PRKCZ was hypermethylated and associated with reduced gene expression. However, the relationship between PRKCZ methylation and the biological behaviors of hepatocellular carcinoma cells has not been thoroughly studied [48]. Thus, it is the first study to focus on the role of PRKCZ methylation in cancers, and a detailed mechanistic study is provided.

HPV-mediated cancer development is inseparable from the continuous expression of the viral E6 and E7 oncogenes [49]. In addition to the role of E6 in proteolytic degradation of tumor suppressor p53 and E7 in interference with the activity of tumor suppressor pRb [50], the deregulation of E6 and E7 in other carcinogenic mechanisms is also important. To verify the effect of E6 and E7 on PRKCZ methylation, we used E6 and E7 lentivirus plasmids to transfect the HPV- HNSCC cell line and E6 or E7 siRNA to transfect the HPV+ HNSCC cell line. Our results revealed that the expression of HPV E6 and E7, especially E6, resulted in PRKCZ promoter hypermethylation by stimulating DNMT1 expression, which led to decreased expression of PRKCZ in HPV+ HNSCC cells. Additionally, consistent with our results, HPV E6 was found to increase DNMT activity and expression, ultimately resulting in impaired E-cadherin expression in cervical cancer [51].

In our study, we used two methods to inhibit PRKCZ expression, including siRNA and a pseudosubstrate inhibitor, which is a specific and efficient inhibitor of PKC-ζ [24]. However, some studies have proposed that its specificity may not be attributed to the high degree of sequence homology between PKC-ζ and PKC-ι [52]. Thus, in our study, the application of a pseudosubstrate inhibitor was employed. Our results suggested that PRKCZ acted as a potential tumor promoter for HPV+ HNSCC. Inhibiting PRKCZ expression led to impaired proliferation, migration, and invasion abilities, and increased the apoptosis rate of HPV+ HNSCC cells. In addition, blocking PRKCZ expression delayed the tumor growth of HPV16-E6/E7 transgenic mice. These results were consistent with previous studies in prostate cancer, lung cancer, and HNSCC, where PRKCZ expression was higher than that in respective normal tissues. After knocking down PRKCZ expression via siRNA, prostate cancer cells presented impaired proliferation, migration, and tumorigenesis, which resulted from changed voltage-gated K+ channel activity [53]. The application of a PKC-ζ specific myristoylated pseudosubstrate inhibitor impaired the chemotactic capacity of non-small cell lung cancer cells [54]. PRKCZ acted as a potent tumor promoter in HNSCC by influencing DNA synthesis [55], holoprotein integrity and telomerase activity [56], and metastatic activity [57]. However, the role of PKC-ζ in oncogenesis remains controversial. Some studies have proposed the exact opposite viewpoints. They suggested that PRKCZ acts as a potential tumor suppressor in intestinal tumors, prostate cancer, and lung cancer via involvement in metabolism reprogramming [58], suppressing the polarization of macrophages to the M2 phenotype [59], and inhibiting EMT [60]. The reasons for the differences among these results remain unknown and must be solved.

Cdc42 is a Rho GTPase responsible for the dynamic reorganization of the actin filament system [61]. Cdc42 plays a pivotal role in migration, invasion, and EMT via the regulation of cytoskeletal and microtubule dynamics [62]. For example, Cdc42 is overexpressed and activated in hepatocellular carcinoma, correlating with reduced E-cadherin expression and an enhanced EMT phenotype [63]. In addition, Cdc42 functions together with Par-3, atypical PKC, and Par-6 in cell polarity and asymmetric cell division [64]. In nasopharyngeal cancer cells, Cdc42 modulates PKC-ζ expression to control telomerase activity. Here, our results suggest that PRKCZ mediates the level of Cdc42 to regulate EMT in HPV+ HNSCC cells. In addition, HPV18 E6 activates the small GTPase Rac1 and subsequently contributes to the E6 stimulation of NFκB in cervical cancer [65]. Depletion of PKC-ζ inhibits nuclear localization of NFκB-p65, inhibiting EMT, invasion, and metastatic progression [66]. Hence, NFκB might be necessary for PRKCZ mediating Cdc42 to regulate EMT in HPV+ HNSCC cells. More should be performed in the future.

In our HPV16-E6/E7 transgenic mouse model of tongue tumors, we found that tumor lesions in the tongue were much smaller in the inhibitor-treated and siRNA-treated groups compared with the DEPC-treated group, which might be due to decreased Cdc42 expression and increased E-cad expression. In our animal experiments, we did not demonstrate whether demethylation therapy with PRKCZ was effective in suppressing tumor growth because this function had been reported by Biktasova A et al. These researchers proposed that the application of 5-aza could inhibit HPV+ HNSCC tumor growth and mouse blood vessel invasion by repressing the expression and activity of matrix metalloproteinases [67]. Therefore, we focused on the effect of blocking PRKCZ treatment rather than repetitively testing the effect of demethylation therapy. However, it will be interesting to explore the effect of combination therapy between blocking PRKCZ and demethylation therapy.

Overall, we illustrated for the first time that PRKCZ was hypermethylated in HPV+ HNSCC. In addition, PRKCZ acts as a tumor promotor that induces HPV+ HNSCC cell proliferation, invasion, and migration and prevents apoptosis. Mechanistically, HPV E6 might promote the expression of DNMT1 to induce PRKCZ hypermethylation, and PRKCZ increased Cdc42 levels to promote EMT in HPV+ HNSCC cells. These results indicated that PRKCZ hypermethylation might represent one possible reason why HPV+ HNSCC has a better prognosis and that blocking PRKCZ might be a new strategy for HPV+ HNSCC.

## Figures and Tables

**Figure 1 cancers-14-04151-f001:**
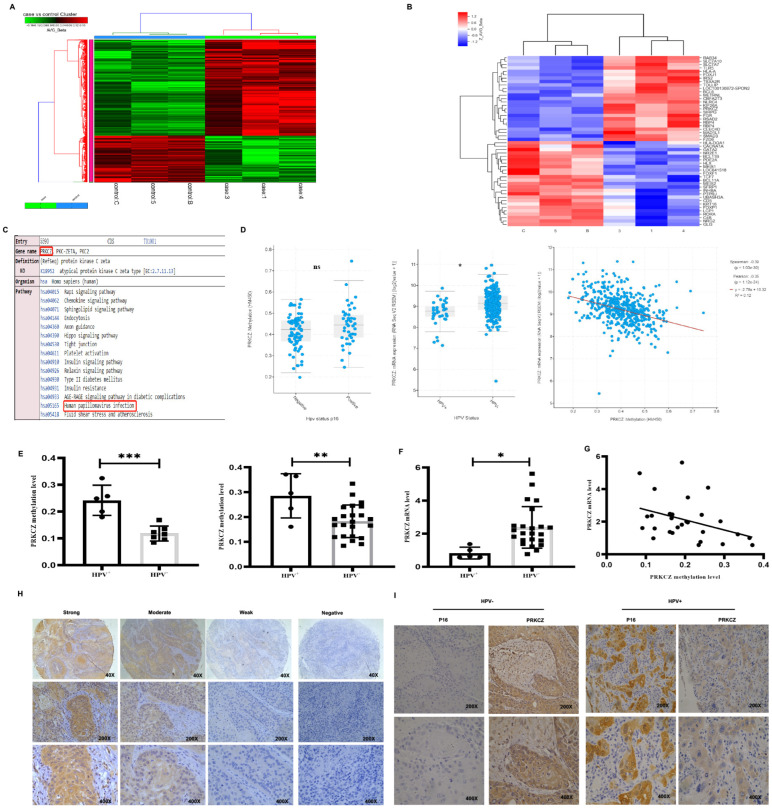
PRKCZ is hypermethylated in HPV+ HNSCC. (**A**) Cluster analysis heat map of different hypermethylated and hypomethylated genes in HPV+ HNSCC and HPV- HNSCC. (**B**) Cluster analysis heat map of different Top 25 significantly hypermethylated and 25 obviously hypomethylated genes in HPV+ HNSCC, compared with HPV- HNSCC. (**C**) KEGG enrichment analysis of PRKCZ in Illumina methylation chip and its role in HPV infection. C, 5, and B represented HPV-negative HNSCC patients, while 1, 3, and 4 represented HPV-positive HNSCC patients. (**D**) PRKCZ methylation degree and mRNA expression level in HPV+ and HPV- HNSCC in TCGA database. PRKCZ methylation degree is inversely associated with mRNA expression level (r = −0.34). (**E**) PRKCZ was hypermethylated in HPV+ HNSCC paraffin-embedded tissues and fresh tissues, compared with HPV- samples. The left represented paraffin-embedded tissues, and the right represented fresh tissues. (**F**) PRKCZ mRNA expression in HPV+ HNSCC fresh tissues was lower than that in HPV- cases. (**G**) The negative correlation between PRKCZ methylation degree and mRNA level in fresh tissues (r = −0.3788, *p* = 0.0469). (**H**) Representative IHC pictures of PRKCZ in HNSCC with strong positive, medium positive, weak positive, and negative staining. (**I**) Representative IHC pictures of PRKCZ in HPV+/HPV-HNSCC. All assays were carried out in triplicate. Results were presented as means ± SD. * *p* < 0.05, ** *p* < 0.01, *** *p* < 0.001.

**Figure 2 cancers-14-04151-f002:**
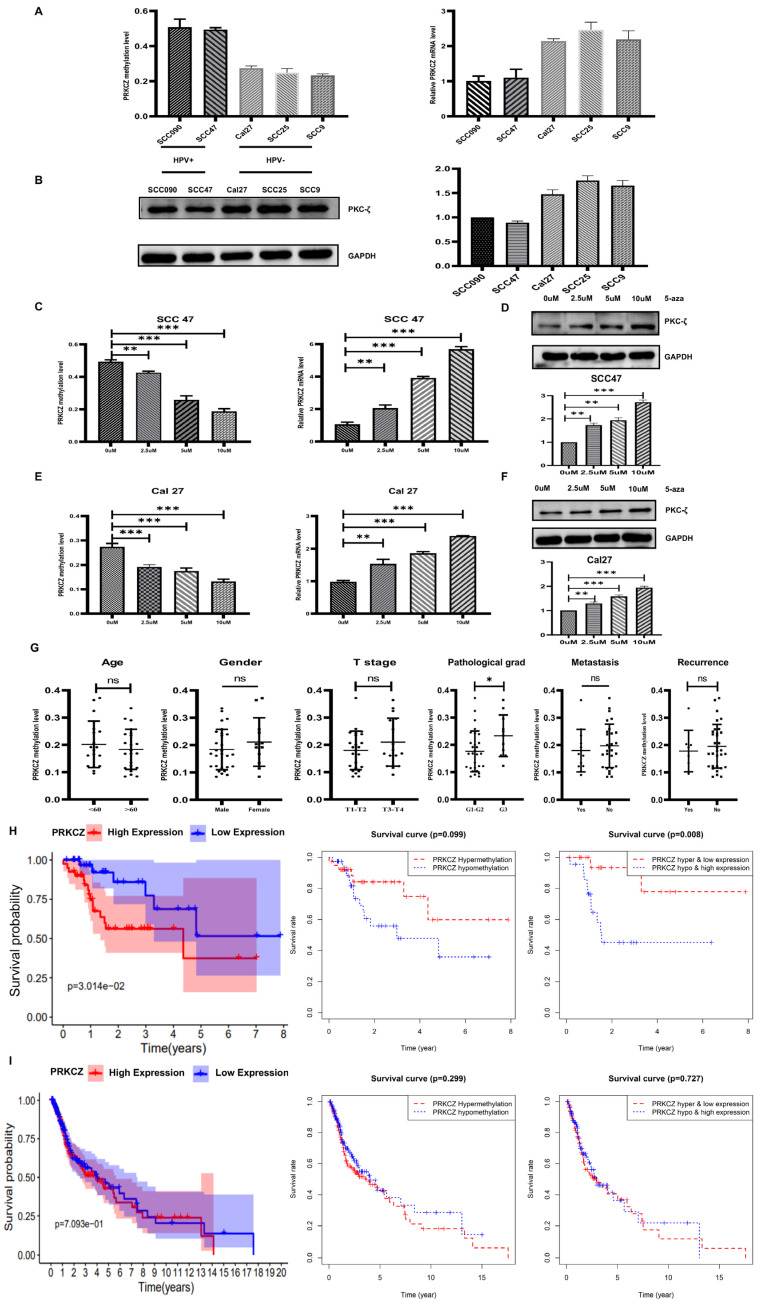
PRKCZ methylation degree in HPV+/HPV- HNSCC cell lines and its clinicopathological significance. (**A**) The PRKCZ methylation degree and mRNA level in HPV+/HPV- HNSCC cell lines. (**B**) The PKC-ζ protein expression and densitometry analysis of WB results in HPV+/HPV- HNSCC cell lines. (**C**) The relationship between 5-aza concentration and methylation degree and mRNA levels of PRKCZ in SCC47. (**D**) The relationship between 5-aza concentration and protein levels of PKC-ζ in SCC47 (include densitometry analysis of WB results). (**E**) The relationship between 5-aza concentration and methylation degree and mRNA levels of PRKCZ in Cal27. (**F**) The relationship between 5-aza concentration and protein levels of PKC-ζ in Cal27 (include densitometry analysis of WB results). (**G**) The predictive value of PRKCZ methylation status for relevant clinical and pathological parameters. *p* values of sex, age, T stage, lymph node metastasis, tumor recurrence, and pathological grading were 0.3138, 0.4679, 0.2365, 0.5322, 0.5827, and 0.0386, respectively. (**H**) Kaplan–Meier survival analysis of HPV+ HNSCC patients with high and low PRKCZ expression, PRKCZ hypermethylation and hypomethylation; PRKCZ hypermethylation and low expression and PRKCZ hypomethylation and high expression. (**I**) Kaplan–Meier survival analysis of HPV- HNSCC patients with high and low PRKCZ expression, PRKCZ hypermethylation and hypomethylation; PRKCZ hypermethylation and low expression and PRKCZ hypomethylation and high expression. All assays were carried out in triplicate. Results were presented as means ± SD. ns > 0.05, * *p* < 0.05, ** *p* < 0.01, *** *p* < 0.001.

**Figure 3 cancers-14-04151-f003:**
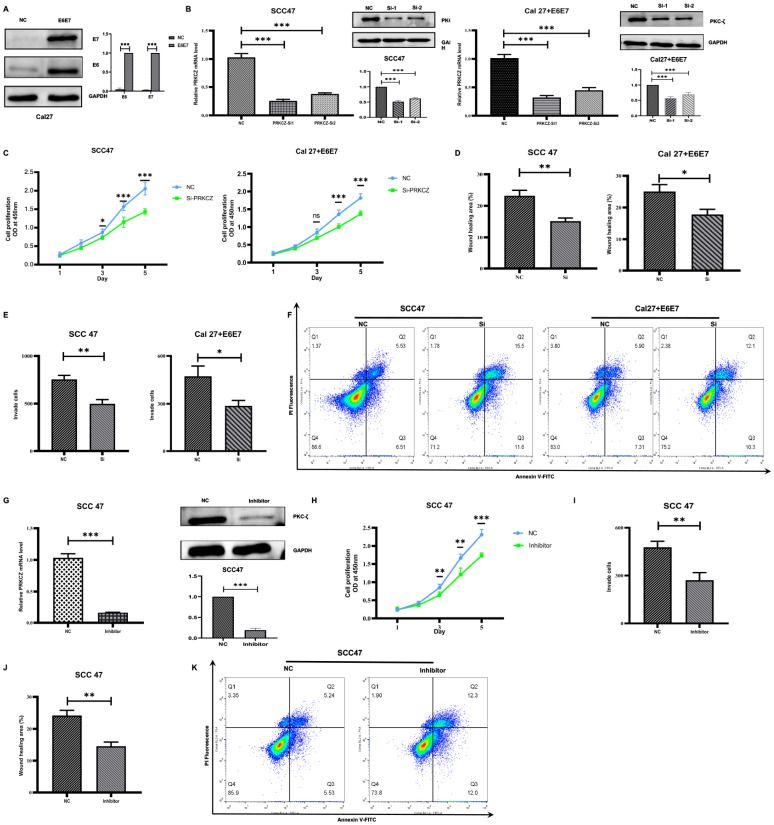
The effect of PRKCZ on biological behaviors of HPV+ HNSCC cell lines. (**A**) Transfection efficiency of E6 and E7 lentivirus plasmids (include densitometry analysis of WB results). (**B**) Transfection efficiency of PRKCZ siRNAs (include densitometry analysis of WB results). (**C**) Silencing PRKCZ effects on proliferation abilities of HPV+ HNSCC cell lines. (**D**) Silencing PRKCZ effects on migration abilities of HPV+ HNSCC cell lines. (**E**) Silencing PRKCZ effects on invasion abilities of HPV+ HNSCC cell lines. (**F**) Silencing PRKCZ effect on cell apoptosis of HPV+ HNSCC cell lines. (**G**) Inhibition efficiency of PKC-ζ pseudosubstrate inhibitor (include densitometry analysis of WB results). (**H**) The role of PKC-ζ pseudosubstrate inhibitor on proliferation abilities of HPV+ HNSCC cell lines. (**I**) The role of PKC-ζ pseudosubstrate inhibitor on migration abilities of HPV+ HNSCC cell lines. (**J**) The role of PKC-ζ pseudosubstrate inhibitor on invasion abilities of HPV+ HNSCC cell lines. (**K**) The role of PKC-ζ pseudosubstrate inhibitor on cell apoptosis of HPV+ HNSCC cell lines. All assays were carried out in triplicate. Results were presented as means ± SD. ns > 0.05, * *p* < 0.05, ** *p* < 0.01, *** *p* < 0.001.

**Figure 4 cancers-14-04151-f004:**
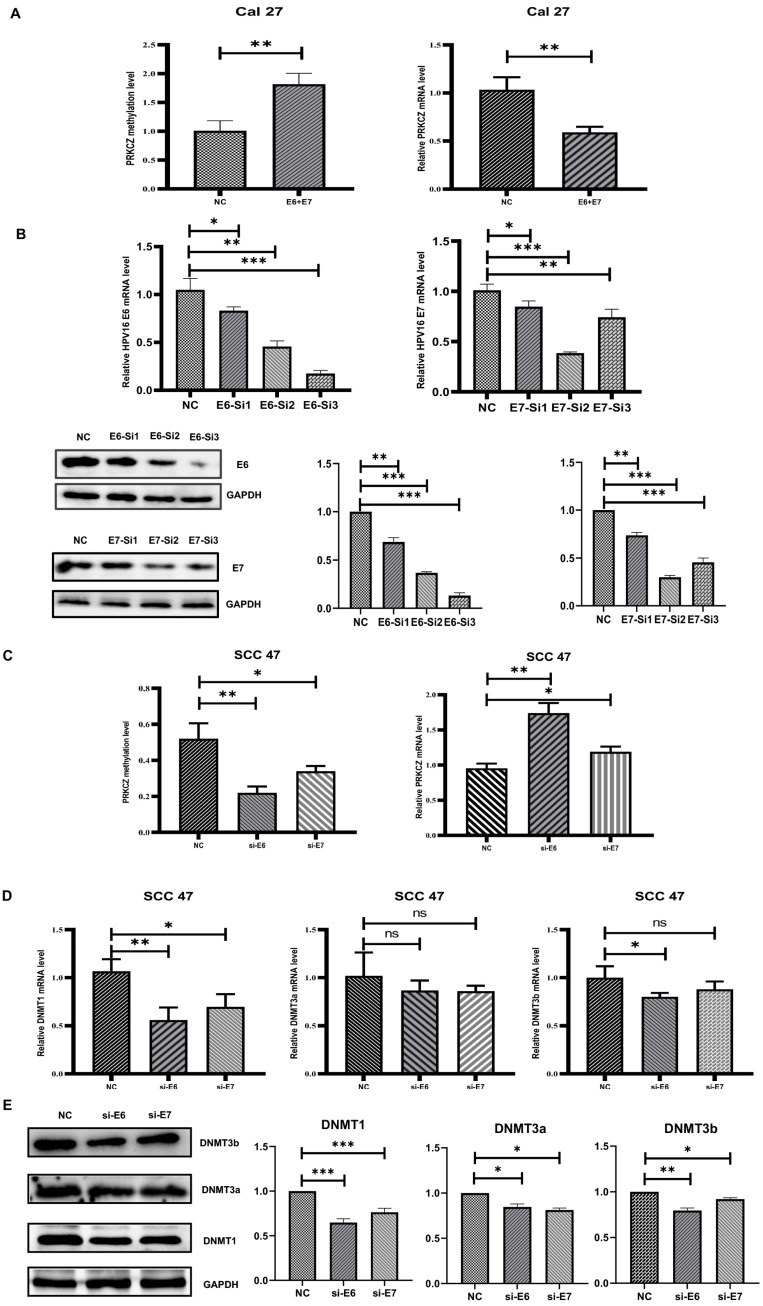
DNMT1 was necessary in HPV-E6 regulating PRKCZ hypermethylation. (**A**) The methylation degree and mRNA level of PRKCZ in Cal27 after transfecting with E6 and E7 lentivirus plasmids. (**B**) Transfection efficiency of E6 and E7 siRNAs. The two pictures in the first row were about qPCR results. The three pictures in the second row were about WB results (include densitometry analysis). (**C**) The methylation degree and mRNA level of PRKCZ in SCC47 after silencing E6 or E7 expression. (**D**) The mRNA expressions of DNMT1, DNMT3a and DNMT3b in SCC47 after silencing E6 or E7 expression. (**E**) The protein expressions of DNMT1, DNMT3a, and DNMT3b in SCC47 after silencing E6 or E7 expression (include densitometry analysis of WB results). All assays were carried out in triplicate. Results were presented as means ± SD. ns > 0.05, * *p* < 0.05, ** *p* < 0.01, *** *p* < 0.001.

**Figure 5 cancers-14-04151-f005:**
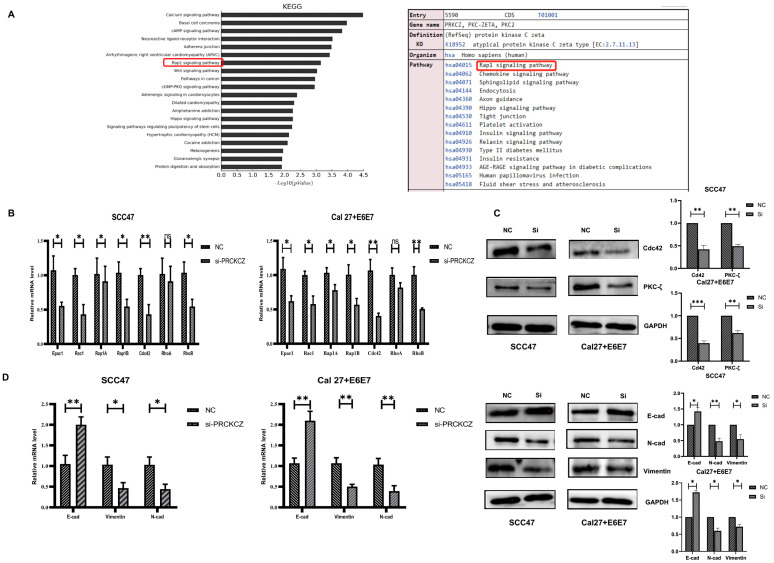
PRKCZ regulated Cdc42 to mediate EMT of HPV+ HNSCC cells. (**A**) The top 20 KEGG pathways for abnormal methylation gene enrichment in Illumina methylation chip and PRKCZ role in Rap signaling pathway. (**B**) The mRNA expression changes of Epac1, Rac1, Rap1A, Rap1B, Cdc42, RhoA, and RhoB in HPV+ HNSCC cells after PRKCZ silencing. (**C**) The protein expression changes of Cdc42 in HPV+ HNSCC cells after PRKCZ silencing (include densitometry analysis of WB results). (**D**) The mRNA and protein expression (include densitometry analysis of WB results) changes of E-cadherin, N-cadherin, and Vimentin in HPV+ HNSCC cells after PRKCZ silencing. All assays were carried out in triplicate. Results were presented as means ± SD. ns > 0.05, * *p* < 0.05, ** *p* < 0.01, *** *p* < 0.001.

**Figure 6 cancers-14-04151-f006:**
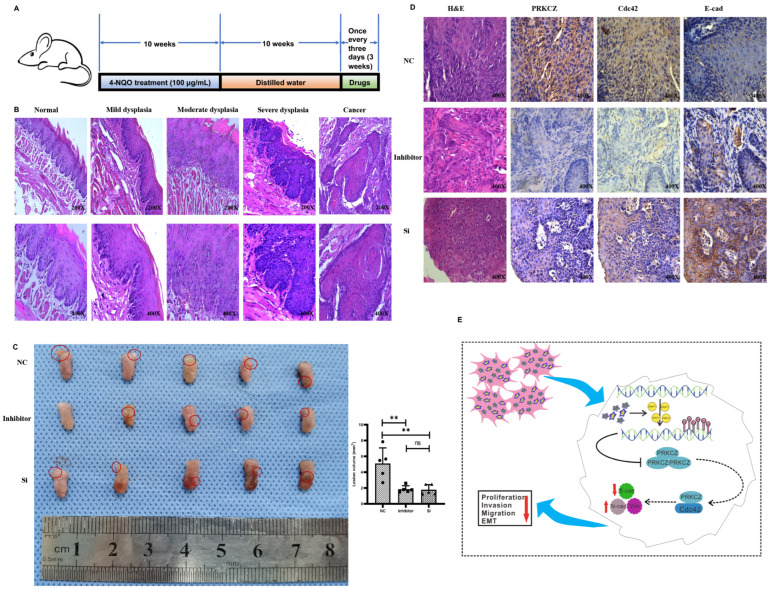
Blocking PRKCZ delayed the tumor growth of HPV16-E6/E7 transgenic mice. (**A**) Schematic diagram illustrated the mice feeding process. (**B**) Representative HE staining pictures of 4-NQO treated transgenic mice. (**C**) The tumor general views and volumes of three groups. (**D**) Representative IHC pictures of PRKCZ, Cdc42, and E-cadherin in three groups. Quantitative analysis based on triplicate experiments. (**E**) Schematic diagram illustrated that the role of PRKCZ hypermethylation in HPV+ HNSCC progression. Results were presented as means ± SD. ns > 0.05, ** *p* < 0.01.

## Data Availability

The data that support the findings of this study are available from the corresponding author upon reasonable request.

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
