# Peer review of "Hypermethylation of PRKCZ Regulated by E6 Inhibits Invasion and EMT via Cdc42 in HPV-Related Head and Neck Squamous Cell Carcinoma"

_cancers, 2022, doi:10.3390/cancers14174151_

Round 1
Reviewer 1 Report
In this manuscript, Wang et al. demonstrate that the PRKCZ gene is specifically hyper-methylated in HPV+ HNSCC and demonstrate the mechanistic implications of this in these cancers. Whilst potentially of interest to the field, several issues with background, interpretation and experimental data should be addressed before publication.
- There are almost triple the number of cases/deaths of HNSCC per year than the authors suggest - please update the reference (see Chow et al., NEJM, 2018 for reference)
- More background on how HPV causes cancer would be useful here (see recent review Scarth et al., J Gen Virol, 2021)
- A lot of the data cited here stems from the TCGA HNSCC paper - this should be cited
- Figure 1 - resolution of C and D is not acceptable - please increase resolution and ensure text is legible in the whole figure (A-I) without high zoom
- Figure 2 - although two representative cell lines is sufficient for mechanistic analysis, in 2A and B, more HPV- and HPV+ cell lines should be included. Furthermore, this should include protein analysis of PKC-ζ
- Throughout the text, the authors should refer to protein analysis of PKC-ζ, not PRKCZ, as they have in the figures
- Figure 3A - the authors should show the data from Figure 4A and B in Figure 3 to demonstrate the effect of E6/E7 in this methylation and repression. Furthermore, protein expression must also be analysed and the depletion of E6/E7 should be shown
- Cal27 + E6/E7 cells can NOT be referred to as HPV+ HNSCC cells, as they are not
- Figure 4C -protein expression of DNMT protein must be analysed must also be analysed
- Figure 5A - resolution is not acceptable - please increase
- Figure 6E - resolution is not acceptable - please increase
- HPV activates Rac1/NFkB signalling (Morgan et al., Plos Pathogens, 2019). This data should be discussed due to the implications of GTPase signalling offered here, particularly as PKC-ζ and cdc42 can also activate NFkB
Author Response
Responses
In this manuscript, Wang et al. demonstrate that the PRKCZ gene is specifically hyper-methylated in HPV+ HNSCC and demonstrate the mechanistic implications of this in these cancers. Whilst potentially of interest to the field, several issues with background, interpretation and experimental data should be addressed before publication.
- There are almost triple the number of cases/deaths of HNSCC per year than the authors suggest - please update the reference (see Chow et al., NEJM, 2018 for reference)
【Author response】 Thank you for your professional comment! We have updated the data in 2018 according to Chow et al. We changed the number of cases/deaths of HNSCC per year in the first paragraph of the “Introduction” part. And the details are as follows: (See Page 1 Para 1 Line 1-4)
Head and neck squamous cell carcinoma (HNSCC) is the most common malignant tumor stemming from oral cavity with about 300,000 new cases and 145,000 death cases worldwide every year → Head and neck squamous cell carcinoma (HNSCC) is the most common malignant tumor stemming from the oral cavity with approximately 890,000 new cases and 450,000 deaths occurring worldwide in 2018[1, 2].
Reference
- Chow, L. Q. M. Head and Neck Cancer. N Engl J Med 2020, 382, 1: 60-72. https://doi.org/10.1056/NEJMra1715715
- Bray, F., J. Ferlay, I. Soerjomataram, R. L. Siegel, L. A. Torre, and A. Jemal. Global Cancer Statistics 2018: Globocan Estimates of Incidence and Mortality Worldwide for 36 Cancers in 185 Countries. CA Cancer J Clin 2018, 68, 6: 394-424. https://doi.org/10.3322/caac.21492
- More background on how HPV causes cancer would be useful here (see recent review Scarth et al., J Gen Virol, 2021)
【Author response】 Thank you for your constructive suggestion! We have added some researches about how HPV causes cancer in the first paragraph of the “Introduction” part. And the details are as follows: (See Page 1 Para 1 Line 12-19)
The oncoproteins E5, E6 and E7 play vital nonnegligible roles in the carcinogenesis of HPV16[7]. E7 binds to and induces degradation of the retinoblastoma protein (pRb) to dysregulate the G1/S-phase transition, while E6 targets the degradation of the p53 tumor suppressor protein to disrupt pro-apoptotic signaling. Besides, E5 could drive cell proliferation. Thus, a persistent HPV infection makes cancers much more likely to develop and progress[8].
Reference
- Scarth, J. A., M. R. Patterson, E. L. Morgan, and A. Macdonald. The Human Papillomavirus Oncoproteins: A Review of the Host Pathways Targeted on the Road to Transformation. J Gen Virol 2021, 102, 3 https://doi.org/10.1099/jgv.0.001540
- Moody, C. A., and L. A. Laimins. Human Papillomavirus Oncoproteins: Pathways to Transformation. Nat Rev Cancer 2010, 10, 8: 550-60. https://doi.org/10.1038/nrc2886
- A lot of the data cited here stems from the TCGA HNSCC paper - this should be cited
【Author response】Thank you for your advice! We have added the references about the data from TCGA. The label numbers of these references are 28 and 29.
Reference
- Cerami, E., J. Gao, U. Dogrusoz, B. E. Gross, S. O. Sumer, B. A. Aksoy, A. Jacobsen, C. J. Byrne, M. L. Heuer, E. Larsson, Y. Antipin, B. Reva, A. P. Goldberg, C. Sander, and N. Schultz. The Cbio Cancer Genomics Portal: An Open Platform for Exploring Multidimensional Cancer Genomics Data. Cancer Discov 2012, 2, 5: 401-4. https://doi.org/10.1158/2159-8290.CD-12-0095
- Gao, J., B. A. Aksoy, U. Dogrusoz, G. Dresdner, B. Gross, S. O. Sumer, Y. Sun, A. Jacobsen, R. Sinha, E. Larsson, E. Cerami, C. Sander, and N. Schultz. Integrative Analysis of Complex Cancer Genomics and Clinical Profiles Using the Cbioportal. Sci Signal 2013, 6, 269: pl1. https://doi.org/10.1126/scisignal.2004088
- Figure 1 - resolution of C and D is not acceptable - please increase resolution and ensure text is legible in the whole figure (A-I) without high zoom
【Author response】Thank you for your advice! We have reuploaded the Figure 1 with higher resolution. Besides, due to the right figure of Figure 1C could not be clearly placed, we have put it into the supplementary materials (Figure S1).
- Figure 2 - although two representative cell lines is sufficient for mechanistic analysis, in 2A and B, more HPV- and HPV+ cell lines should be included. Furthermore, this should include protein analysis of PKC-ζ
【Author response】 Thank you for your valuable proposal! We have added 2 HPV+ HNSCC cell lines (SCC47 and SCC090), 3 HPV- HNSCC cell lines (Cal27, SCC25 and SCC9) to repeat our data. The qMSP, qPCR and WB experiments were conducted. And the results have been put into Figure 2A and 2B of the revised manuscript.
(A) The PRKCZ methylation degree and mRNA level in HPV+/HPV- HNSCC cell lines. (B) The PKC-ζ protein expression and densitometry analysis of WB results in HPV+/HPV- HNSCC cell lines.
- Throughout the text, the authors should refer to protein analysis of PKC-ζ, not PRKCZ, as they have in the figures
【Author response】 We appreciate your valuable suggestion! We have changed the protein expression of PRKCZ into the protein expression of PKC-ζ at the appropriate places in the revised manuscript.
- Figure 3A - the authors should show the data from Figure 4A and B in Figure 3 to demonstrate the effect of E6/E7 in this methylation and repression. Furthermore, protein expression must also be analysed and the depletion of E6/E7 should be shown
【Author response】 Thank you for your professional advice! Figure 3 is the results about PRKCZ effect on the biological behaviors of HPV+ HNSCC cells, and Figure 4 is the results about how HPV influenced PRKCZ methylation. Figure 3 has 11 (A-K) results, whereas Figure 4 has 5 (A-E) results. Thus, we put the results of the effect of E6/E7 in methylation and repression into Figure 4, to avoid Figure 3 more crowded.
And the WB results and protein expression of the E6 and E7 expression in Cal27+E6E7 cells was shown in Figure 3A. In addition, the qPCR and WB experiments (including protein expression analysis) have been conducted, and the results were shown in Figure 4B.
Transfection efficiency of E6 and E7 siRNAs. The two pictures in the first row were about qPCR results. The three pictures in the second row were about WB results (include densitometry analysis).
- Cal27 + E6/E7 cells can NOT be referred to as HPV+ HNSCC cells, as they are not
【Author response】 Thank you for your advice! We have made a change about the name of Cal+E6E7 cell in the “Results 3.3” part. And the details are as follows: (See Page 10 Para 2 Line 2)
we establish a new HPV+ HNSCC cell line, named Cal+E6E7 cell, via transfecting Cal27 cell with E6 and E7 lentivirus plasmid. → we established a new HPV+ HNSCC like cell line termed Cal+E6E7 cell by transfecting Cal27 cells with E6 and E7 lentivirus plasmids.
- Figure 4C -protein expression of DNMT protein must be analysed must also be analysed
【Author response】 Thank you for your professional comment! WB experiments about protein expressions of DNMT proteins have been supplemented, which has been shown in Figure 4D.
(D) The protein expressions of DNMT1, DNMT3a and DNMT3b in SCC47 after silencing E6 or E7 expression (include densitometry analysis of WB results).
- Figure 5A - resolution is not acceptable - please increase
【Author response】Thank you for your advice! We have reuploaded the Figure 5 with higher resolution. Besides, due to the right figure of Figure 5A could not be clearly placed, we have put it into the supplementary materials (See Figure S6).
- Figure 6E - resolution is not acceptable - please increase
【Author response】Thank you for your advice! We have reuploaded the Figure 6 with higher resolution (See Figure 6).
- HPV activates Rac1/NFkB signalling (Morgan et al., Plos Pathogens, 2019). This data should be discussed due to the implications of GTPase signalling offered here, particularly as PKC-ζ and cdc42 can also activate NFkB
【Author response】 Thank you for your professional and valuable suggestions! We have discussed the reference you mentioned in the sixth paragraph of “Discussion” part. And the details are as follows: (See Page 17 Para 4 Line 9-12)
Here, our results suggested that PRKCZ mediates the level of Cdc42 to regulate EMT in HPV+ HNSCC cells. In addition, HPV18 E6 activates the small GTPase Rac1 and subsequently contributes to the E6 stimulation of NFκB in cervical cancer [65]. Depletion of PKC-ζ inhibits nuclear localization of NFκB-p65, inhibiting EMT, invasion and metastatic progression[66]. Hence, NFκB might be necessary in PRKCZ mediating Cdc42 to regulate EMT in HPV+ HNSCC cells. More should be done in the future.
Reference
- Morgan, E. L., and A. Macdonald. Autocrine Stat3 Activation in Hpv Positive Cervical Cancer through a Virus-Driven Rac1-Nfkappab-Il-6 Signalling Axis. PLoS Pathog 2019, 15, 6: e1007835. https://doi.org/10.1371/journal.ppat.1007835
- Paul, A., M. Danley, B. Saha, O. Tawfik, and S. Paul. Pkczeta Promotes Breast Cancer Invasion by Regulating Expression of E-Cadherin and Zonula Occludens-1 (Zo-1) Via Nfkappab-P65. Sci Rep 2015, 5: 12520. https://doi.org/10.1038/srep12520
Thank you for your opinion. Our manuscript has been edited for proper English language, grammar, punctuation, spelling, and overall style by one or more of the highly qualified native English speaking editors at AJE (American Journal Experts). The main changes are as follows:
(1) Page 2 Para 1 Line 4-8: Although the treatment technologies and means are constantly improving, the 5-year survival rate of HNSCC patients is still not satisfactory, only about 50%. → Despite constantly improvements in treatments, the 5-year survival rate of HNSCC patients remains unsatisfactory at only approximately 50%.
(2) Page 2 Para 3 Line 3: , where → in which
(3) Page 6 Pare 2 Line 6-8: it did not have statistical difference of the methylation degree of PRKCZ between HPV+ HNSCC and HPV- cases. → no significant difference in the methylation degree of PRKCZ was noted between HPV+ HNSCC and HPV- cases.
(4) Page 6 Pare 2 Line 16: added “the use of different”; deleted “of”
(5) Page 6 Pare 2 Line 18: found → localized
(6) Page 8 Pare 1 Line 7-8: With the increased concentration of 5-aza → As the concentration of 5-aza increased
(7) Page 10 Pare 1 Line 18-22: There is no difference between PRKCZ methylation status and relevant clinical and pathological parameters in HPV+ patients and the same as in HPV- patients. → No differences between PRKCZ methylation status and relevant clinical and pathological parameters were noted in HPV+ patients and HPV- patients.
(8) Page 10 Pare 2 Line 3-5: The result of Western blot suggested the expression of E6 and E7 in new cell lines, → The Western blot results revealed E6 and E7 expression in the new cell line,
(9) Page 10 Pare 2 Line 9-11: Western blot experiments also proved two pairs of siRNA the silencing effects on PRKCZ protein expression. → Western blot experiments also demonstrated the silencing effects of both pairs of siRNAs on PKC-ζ protein expression.
(10) Page 11 Pare 1 Line 2-8: Besides, the wound healing areas (Figure 3I, Figure S4) and in-vade cells number (Figure 3J, Figure S4) of SCC47 treated with pseudosubstrate inhibi-tor were 14.571% and 271, respectively, while those of normal SCC47 were 24.174% and 478, respectively. → In addition, the percentage of wound healing areas (Figure 3I, Figure S5) and number of invading SCC47 cells (Figure 3J, Figure S5) treated with the pseudosubstrate inhibi-tor were 14.571% and 271, respectively, while those of normal SCC47 cells were 24.174% and 478, respectively.
(11) Page 12 Pare 1 Line 5-10: the methylation degree of PRKCZ was increased, while the mRNA level was decreased. To further prove which one was more powerful, E6 or E7, the levels of E6 and E7 in SCC47 were inhibited via respective siRNA. → PRKCZ methylation levels were increased, whereas PRKCZ mRNA levels decreased (Figure 4A). To further determine whether E6 or E7 was more powerful, E6 and E7 lev-els in SCC47 were inhibited using respective siRNAs.
(12) Page 12 Pare 2 Line 2-4: we speculated whether HPV mediated PKRCZ methylation via regulating the expressions of DNMTs family. → we hypothesized that HPV mediates PKRCZ methylation by regulating the expression of the DNMT family.
(13) Page 12 Pare 2 Line 7-9: the level of DNMT1 was decreased, while the levels change of DNMT3a and DNMT3b was not obvious. → DNMT1 levels decreased, whereas changes in the levels of DNMT3a and DNMT3b were not obvious.
(14) Page 13 Pare 1 Line 8-11: we speculated whether PRKCZ regulated Rap signaling pathway to mediate EMT, finally influencing migration and invasion of HPV+ HNSCC cells. → we hypothesized that PRKCZ regulated the Rap signaling pathway to mediate EMT, ultimately influencing HPV+ HNSCC cell migration and invasion.
(15) Page 14 Pare 1 Line 1-6: The results of qPCR suggested that after PRKCZ silenced, the mRNA expressions of Epac1, Rac1, Rap1A, Rap1B, Cdc42 and RhoB were decreased, while the change of RhoA had no statistical difference. Besides, the mRNA level of Cdc42 changed most dramatically. → The qPCR results suggested that after PRKCZ silencing, Epac1, Rac1, Rap1A, Rap1B, Cdc42 and RhoB mRNA expression decreased, whereas the change in RhoA was not significantly different. In addition, Cdc42 mRNA levels changed most dramatically.
(16) Page 15 Pare 1 Line 9-10: and following with distilled water for another 10 weeks →followed by distilled water for an additional 10 weeks
(17) Page 16 Pare 2 Line 5-7: while HPV+ counterpart seems to have more maintenance of global methylation, and preform → whereas HPV+ counterparts seem to have increased maintenance of global methylation and exhibit
(18) Page 16 Pare 2 Line 17-19: our insufficient clinical specimens might be a reason. → the insufficient number of clinical specimens in our study might explain the difference.
(19) Page 16 Pare 3 Line 5-9: the methylation degree difference from TCGA data has no statistical difference, which may due to detection method discrepancy of PRKCZ methylation degree and HPV status. → the difference in the level of methylation observed in TCGA data was not significant, which may be due to a discrepancy in the PRKCZ methylation degree and HPV status.
(20) Page 16 Pare 3 Line 11-13: There is, as we know, just one research reporting its role in cancers. → However, to the best of our knowledge, only one study has reported the role of PRKCZ in cancers.
(21) Page 17 Pare 3 Line 3-6: its specificity may be not what it was supposed to be, due to the high degree of sequence homology between PKC-ζ and PKC-ι. → its specificity may not be attributed to the high degree of sequence homology between PKC-ζ and PKC-ι.
(22) Page 17 Pare 3 Line 19-22: However, what role does PKC-ζ play in onco-genesis is still controversial. There were some researches proposing the exact opposite viewpoints. → However, the role of PKC-ζ in oncogenesis remains controversial. Some studies have proposed the exact opposite viewpoints.
(23) Page 17 Pare 3 Line 25-27: What accounts for the differences among these results is still betwixt and be-tween, and urgent to be solved. → The reasons for the differences among these results remains unknown and must be solved.
(24) Page 18 Pare 2 Line 2-5: and as a tumor promotor which was in favor of proliferation, invasion and migration potential, and not conducive to apoptosis of HPV+ HNSCC cells. → In addition, PRKCZ acts as a tumor promotor that induces HPV+ HNSCC cell proliferation, invasion and migration and prevents apoptosis.

Reviewer 2 Report
The current study, "Hypermethylation of PRKCZ Regulated by E6/DNMT1 Inhibits Invasion and EMT via Cdc42 in HPV-Related Head and Neck Squamous Cell Carcinoma," by Wang et al., shows that silencing PRKCZ damaged the malignant capacity of HPV+ HNSCC cells, delayed tumor growth in HPV16-E6/E7 transgenic mice, and inhibited the epithelial-mesenchymal transition (EMT). They concluded that hypermethylated PRKCZ may induce EMT via Cdc42 and act as a potent tumor promoter in HPV+ HNSCC. The findings of this study could be very useful in the future for treating HPV+ HNSCC patients with a new strategy that involves blocking PRKCZ. Overall, the manuscript is well written, nicely presented, and covers the majority of the advanced knowledges; however, there are some questions that should be addressed in order to improve the manuscript's quality.
- In the introduction part, the author stated that the reason why HPV+ HNSCC has a better prognosis is still unknown. However, many arguments or content that describe that HPV+ HNSCC shares a better prognosis are already published/available and should be cited and discussed.
- It would be fantastic if the author could use their bioinformatics skills to correlate their findings in the context of PRKCZ gene expression in HPV+ and HPV- HNSCC from the available database i.e. The Cancer Genome Atlas (TCGA) database.
- The authors demonstrate that blocking PRKCZ delays tumor growth in HPV16-E6/E7 transgenic mice and PRKCZ could delay 4NQO-induced tongue tumor growth, but the mice experiments are not convincing and do not appear to reduce tumor size. It would be fantastic if the author could use the syngeneic mice model to replicate the similar finding by measuring tumor growth on the mice's back flank.
- It would be great if the author could clearly mention in one or two lines about both HNSCC models, i.e. HPV16-E6/E7 (HPV+) and 4NQO-indued (HPV-) models.
Author Response
Responses
The current study, "Hypermethylation of PRKCZ Regulated by E6/DNMT1 Inhibits Invasion and EMT via Cdc42 in HPV-Related Head and Neck Squamous Cell Carcinoma," by Wang et al., shows that silencing PRKCZ damaged the malignant capacity of HPV+ HNSCC cells, delayed tumor growth in HPV16-E6/E7 transgenic mice, and inhibited the epithelial-mesenchymal transition (EMT). They concluded that hypermethylated PRKCZ may induce EMT via Cdc42 and act as a potent tumor promoter in HPV+ HNSCC. The findings of this study could be very useful in the future for treating HPV+ HNSCC patients with a new strategy that involves blocking PRKCZ. Overall, the manuscript is well written, nicely presented, and covers the majority of the advanced knowledges; however, there are some questions that should be addressed in order to improve the manuscript's quality.
- In the introduction part, the author stated that the reason why HPV+ HNSCC has a better prognosis is still unknown. However, many arguments or content that describe that HPV+ HNSCC shares a better prognosis are already published/available and should be cited and discussed.
【Author response】 Thank you for your professional advice! We have read relative references and added some content about why HPV+ HNSCC shares a better prognosis in the first paragraph of “Introduction” part. And the details are as follows:
Compared with HPV- HNSCC, HPV+ HNSCC has distinct genetic and clinicopathological features with better prognosis and sensitivity to radiotherapy and chemotherapy. → Compared with HPV- HNSCC, HPV+ HNSCC has been shown to exhibit a better prognosis and sensitivity to radiotherapy and chemotherapy[9], perform a lower mutational burden, expressed a wild TP53[10, 11], have more tumor infiltrating lymphocytes[12] as well as the less hypoxic[13], however, why HPV+ HNSCC preforms better prognosis and the sensitivity of radiotherapy and chemotherapy still remains unclear.
Reference
- Wang, H. F., S. S. Wang, Y. J. Tang, Y. Chen, M. Zheng, Y. L. Tang, and X. H. Liang. The Double-Edged Sword-How Human Papillomaviruses Interact with Immunity in Head and Neck Cancer. Front Immunol 2019, 10: 653. https://doi.org/10.3389/fimmu.2019.00653
- Stransky, N., A. M. Egloff, A. D. Tward, A. D. Kostic, K. Cibulskis, A. Sivachenko, G. V. Kryukov, M. S. Lawrence, C. Sougnez, A. McKenna, E. Shefler, A. H. Ramos, P. Stojanov, S. L. Carter, D. Voet, M. L. Cortes, D. Auclair, M. F. Berger, G. Saksena, C. Guiducci, R. C. Onofrio, M. Parkin, M. Romkes, J. L. Weissfeld, R. R. Seethala, L. Wang, C. Rangel-Escareno, J. C. Fernandez-Lopez, A. Hidalgo-Miranda, J. Melendez-Zajgla, W. Winckler, K. Ardlie, S. B. Gabriel, M. Meyerson, E. S. Lander, G. Getz, T. R. Golub, L. A. Garraway, and J. R. Grandis. The Mutational Landscape of Head and Neck Squamous Cell Carcinoma. Science 2011, 333, 6046: 1157-60. https://doi.org/10.1126/science.1208130
- Shaikh, H., J. E. McGrath, B. Hughes, J. Xiu, P. Brodskiy, A. Sukari, S. Darabi, C. Ikpeazu, C. Nabhan, W. M. Korn, and T. M. Wise-Draper. Genomic and Molecular Profiling of Human Papillomavirus Associated Head and Neck Squamous Cell Carcinoma Treated with Immune Checkpoint Blockade Compared to Survival Outcomes. Cancers (Basel) 2021, 13, 24 https://doi.org/10.3390/cancers13246309
- Lechien, J. R., I. Seminerio, G. Descamps, Q. Mat, F. Mouawad, S. Hans, M. Julieron, D. Dequanter, T. Vanderhaegen, F. Journe, and S. Saussez. Impact of Hpv Infection on the Immune System in Oropharyngeal and Non-Oropharyngeal Squamous Cell Carcinoma: A Systematic Review. Cells 2019, 8, 9 https://doi.org/10.3390/cells8091061
- Kimple, R. J., M. A. Smith, G. C. Blitzer, A. D. Torres, J. A. Martin, R. Z. Yang, C. R. Peet, L. D. Lorenz, K. P. Nickel, A. J. Klingelhutz, P. F. Lambert, and P. M. Harari. Enhanced Radiation Sensitivity in Hpv-Positive Head and Neck Cancer. Cancer Res 2013, 73, 15: 4791-800. https://doi.org/10.1158/0008-5472.CAN-13-0587
- It would be fantastic if the author could use their bioinformatics skills to correlate their findings in the context of PRKCZ gene expression in HPV+ and HPV- HNSCC from the available database i.e. The Cancer Genome Atlas (TCGA) database.
【Author response】We appreciate your helpful comments! We used bioinformatics skills to assess the prognostic value of PRKCZ methylation and mRNA from TCGA and GEO database. These databases showed that low PRKCZ expression in HPV+ HNSCC patients seem to have a better prognosis, however, HPV+ HNSCC patients with PRKCZ hypermethylation and low expression has a much better survival. In addition, for HPV- HNCSS patients, PRKCZ expression and methylation level have no significant effects on the prognosis. And these results were supplemented in the “Results 3.2” part (See Page 10 Pare 1 Line 10-16) and Figure 2H, 2I in the revised manuscript.
In addition, the data from TCGA database shows that the mRNA degree of PRKCZ was higher in HPV+ HNSCC than that in HPV- cases, and the methylation degree of PRKCZ was lower in HPV+ cases, which was showed in Figure 1D.
(H) HPV+ HNSCC patients with high and low PRKCZ expression, PRKCZ hypermethylation and hypomethylation; PRKCZ hypermethylation & low expression and PRKCZ hypomethylation & high expression.
(I) HPV- HNSCC patients with high and low PRKCZ expression, PRKCZ hypermethylation and hypomethylation; PRKCZ hypermethylation & low expression and PRKCZ hypomethylation & high expression.
- The authors demonstrate that blocking PRKCZ delays tumor growth in HPV16-E6/E7 transgenic mice and PRKCZ could delay 4NQO-induced tongue tumor growth, but the mice experiments are not convincing and do not appear to reduce tumor size. It would be fantastic if the author could use the syngeneic mice model to replicate the similar finding by measuring tumor growth on the mice's back flank.
【Author response】 We appreciate your helpful comments! Figure 6C showed the data of PRKCZ delaying the tongue tumor size of 4NQO-induced HPV16-E6/E7 transgenic mice model. There were several swellings in the mice tongues and the red circle showed the largest tumor. To further show the different tumor sizes of each group, the bar graph is used to sum all tumor volumes via software calculation and statistic analysis has demonstrated blocking PRKCZ delays tumor growth of 4NQO-induced HPV16-E6/E7 transgenic mice model. In the revised manuscript, we have re-uploaded Figure 6 with higher resolution (See Figure 6C).
And the syngeneic mice model you mentioned is a good idea to replicate our experiment. We will consider supplementing this part of the experiment in the future.
- It would be great if the author could clearly mention in one or two lines about both HNSCC models, i.e. HPV16-E6/E7 (HPV+) and 4NQO-indued (HPV-) models.
【Author response】We appreciate your valuable suggestion! 4-NQO supplied in drinking water is widely used as a carcinogen to induce cancers of the oral cavity (HPV-) in rodents, due to the similar histological as well as molecular changes as seen in human oral carcinogenesis[37]. In our experiment, we used 4NQO-induced HPV16-E6/E7 transgenic mice tongue tumor model, which were obtained from Rosa26-E6-E7 constitutive knock-in homozygous C57BL/6 mice. Thus, in the revised manuscript, we have added some contents about these two models in the “Results 3.6” part. (See Page 9 Para 2 Line)
Reference
- Tang, X. H., B. Knudsen, D. Bemis, S. Tickoo, and L. J. Gudas. Oral Cavity and Esophageal Carcinogenesis Modeled in Carcinogen-Treated Mice. Clin Cancer Res 2004, 10, 1 Pt 1: 301-13. https://doi.org/10.1158/1078-0432.ccr-0999-3
Thank you for your opinion. Our manuscript has been edited for proper English language, grammar, punctuation, spelling, and overall style by one or more of the highly qualified native English speaking editors at AJE (American Journal Experts). The main changes are as follows:
(1) Page 2 Para 1 Line 4-8: Although the treatment technologies and means are constantly improving, the 5-year survival rate of HNSCC patients is still not satisfactory, only about 50%. → Despite constantly improvements in treatments, the 5-year survival rate of HNSCC patients remains unsatisfactory at only approximately 50%.
(2) Page 2 Para 3 Line 3: , where → in which
(3) Page 6 Pare 2 Line 6-8: it did not have statistical difference of the methylation degree of PRKCZ between HPV+ HNSCC and HPV- cases. → no significant difference in the methylation degree of PRKCZ was noted between HPV+ HNSCC and HPV- cases.
(4) Page 6 Pare 2 Line 16: added “the use of different”; deleted “of”
(5) Page 6 Pare 2 Line 18: found → localized
(6) Page 8 Pare 1 Line 7-8: With the increased concentration of 5-aza → As the concentration of 5-aza increased
(7) Page 10 Pare 1 Line 18-22: There is no difference between PRKCZ methylation status and relevant clinical and pathological parameters in HPV+ patients and the same as in HPV- patients. → No differences between PRKCZ methylation status and relevant clinical and pathological parameters were noted in HPV+ patients and HPV- patients.
(8) Page 10 Pare 2 Line 3-5: The result of Western blot suggested the expression of E6 and E7 in new cell lines, → The Western blot results revealed E6 and E7 expression in the new cell line,
(9) Page 10 Pare 2 Line 9-11: Western blot experiments also proved two pairs of siRNA the silencing effects on PRKCZ protein expression. → Western blot experiments also demonstrated the silencing effects of both pairs of siRNAs on PKC-ζ protein expression.
(10) Page 11 Pare 1 Line 2-8: Besides, the wound healing areas (Figure 3I, Figure S4) and in-vade cells number (Figure 3J, Figure S4) of SCC47 treated with pseudosubstrate inhibi-tor were 14.571% and 271, respectively, while those of normal SCC47 were 24.174% and 478, respectively. → In addition, the percentage of wound healing areas (Figure 3I, Figure S5) and number of invading SCC47 cells (Figure 3J, Figure S5) treated with the pseudosubstrate inhibi-tor were 14.571% and 271, respectively, while those of normal SCC47 cells were 24.174% and 478, respectively.
(11) Page 12 Pare 1 Line 5-10: the methylation degree of PRKCZ was increased, while the mRNA level was decreased. To further prove which one was more powerful, E6 or E7, the levels of E6 and E7 in SCC47 were inhibited via respective siRNA. → PRKCZ methylation levels were increased, whereas PRKCZ mRNA levels decreased (Figure 4A). To further determine whether E6 or E7 was more powerful, E6 and E7 lev-els in SCC47 were inhibited using respective siRNAs.
(12) Page 12 Pare 2 Line 2-4: we speculated whether HPV mediated PKRCZ methylation via regulating the expressions of DNMTs family. → we hypothesized that HPV mediates PKRCZ methylation by regulating the expression of the DNMT family.
(13) Page 12 Pare 2 Line 7-9: the level of DNMT1 was decreased, while the levels change of DNMT3a and DNMT3b was not obvious. → DNMT1 levels decreased, whereas changes in the levels of DNMT3a and DNMT3b were not obvious.
(14) Page 13 Pare 1 Line 8-11: we speculated whether PRKCZ regulated Rap signaling pathway to mediate EMT, finally influencing migration and invasion of HPV+ HNSCC cells. → we hypothesized that PRKCZ regulated the Rap signaling pathway to mediate EMT, ultimately influencing HPV+ HNSCC cell migration and invasion.
(15) Page 14 Pare 1 Line 1-6: The results of qPCR suggested that after PRKCZ silenced, the mRNA expressions of Epac1, Rac1, Rap1A, Rap1B, Cdc42 and RhoB were decreased, while the change of RhoA had no statistical difference. Besides, the mRNA level of Cdc42 changed most dramatically. → The qPCR results suggested that after PRKCZ silencing, Epac1, Rac1, Rap1A, Rap1B, Cdc42 and RhoB mRNA expression decreased, whereas the change in RhoA was not significantly different. In addition, Cdc42 mRNA levels changed most dramatically.
(16) Page 15 Pare 1 Line 9-10: and following with distilled water for another 10 weeks →followed by distilled water for an additional 10 weeks
(17) Page 16 Pare 2 Line 5-7: while HPV+ counterpart seems to have more maintenance of global methylation, and preform → whereas HPV+ counterparts seem to have increased maintenance of global methylation and exhibit
(18) Page 16 Pare 2 Line 17-19: our insufficient clinical specimens might be a reason. → the insufficient number of clinical specimens in our study might explain the difference.
(19) Page 16 Pare 3 Line 5-9: the methylation degree difference from TCGA data has no statistical difference, which may due to detection method discrepancy of PRKCZ methylation degree and HPV status. → the difference in the level of methylation observed in TCGA data was not significant, which may be due to a discrepancy in the PRKCZ methylation degree and HPV status.
(20) Page 16 Pare 3 Line 11-13: There is, as we know, just one research reporting its role in cancers. → However, to the best of our knowledge, only one study has reported the role of PRKCZ in cancers.
(21) Page 17 Pare 3 Line 3-6: its specificity may be not what it was supposed to be, due to the high degree of sequence homology between PKC-ζ and PKC-ι. → its specificity may not be attributed to the high degree of sequence homology between PKC-ζ and PKC-ι.
(22) Page 17 Pare 3 Line 19-22: However, what role does PKC-ζ play in onco-genesis is still controversial. There were some researches proposing the exact opposite viewpoints. → However, the role of PKC-ζ in oncogenesis remains controversial. Some studies have proposed the exact opposite viewpoints.
(23) Page 17 Pare 3 Line 25-27: What accounts for the differences among these results is still betwixt and be-tween, and urgent to be solved. → The reasons for the differences among these results remains unknown and must be solved.
(24) Page 18 Pare 2 Line 2-5: and as a tumor promotor which was in favor of proliferation, invasion and migration potential, and not conducive to apoptosis of HPV+ HNSCC cells. → In addition, PRKCZ acts as a tumor promotor that induces HPV+ HNSCC cell proliferation, invasion and migration and prevents apoptosis.

Round 2
Reviewer 1 Report
The authors have made good efforts to address all of my comments. I have one remaining issue. The authors suggest that DMNT1 is the main mediator of PRKCZ hypermethylation driven by E6 in HPV+ HNSCC. However, the authors do not experimentally show this by inhibiting DMNT1. Therefore, they should tone down their conclusions and consider altering the title to reflect this.
Author Response
Cover Letter and Responses
Dear Reviewer,
Thank you for your approval of the revisions of our article entitled " Hypermethylation of PRKCZ regulated by E6/DNMT1 inhibits invasion and EMT via Cdc42 in HPV-related head and neck squamous cell carcinoma" (Manuscript ID: cancers-1822871). We have given your latest suggestion serious consideration, carefully responded to this suggestion in this cover letter (see following “Responses” part), and revised the manuscript accordingly. All changes made to the text are highlighted in red color so that they can be identified with ease.
Please contact me if you have any questions. We look forward to hearing from you. Thank you for your time.
Yours sincerely,
Ya-ling Tang MD. PhD
E-mail address: tangyaling@scu.edu.cn
Full postal address: State Key Laboratory of Oral Diseases and National Clinical Research Center for Oral Diseases, West China College of Stomatology, Sichuan University. No.14, Sec.3, Renminnan Road, Chengdu 610041, Sichuan, China
Responses
The authors have made good efforts to address all of my comments. I have one remaining issue. The authors suggest that DMNT1 is the main mediator of PRKCZ hypermethylation driven by E6 in HPV+ HNSCC. However, the authors do not experimentally show this by inhibiting DMNT1. Therefore, they should tone down their conclusions and consider altering the title to reflect this.
【Author response】 We completely agreed with the comment of the reviewer. According to the suggestion, we have changed our title into “Hypermethylation of PRKCZ Regulated by E6 Inhibits Invasion and EMT via Cdc42 in HPV-Related Head and Neck Squamous Cell Carcinoma”. (See Page 1 Title)
Then, we have checked the whole manuscript and changed accordingly.
Abstract: “Mechanistically, HPV promoted DNMT1 expression via E6 to increase PRKCZ methylation.” has been changed into “Mechanistically, HPV might promote DNMT1 expression via E6 to increase PRKCZ methylation.” (See Page 1 Abstract Line 9)
Results: We have changed the title of Result 3.4 into “Methylation Degree of PRKCZ Was Increased by partly DNMT1 via HPV E6”. (See Page 13 Title 3.4) And the conclusion of Result 3.4 has been changed into “Overall, it was demonstrated that in SCC47 cells, HPV at least partly regulated DNMT1 expression through E6, leading to an increase in PRKCZ methylation and a decrease in PRKCZ mRNA expression.” (See Page 13 Para 2 Line 12)
Conclusion: Besides, we have changed our conclusion “Mechanistically, HPV E6 promoted the expression of DNMT1 to induce PRKCZ hypermethylation, and PRKCZ increased Cdc42 levels to promote EMT in HPV+ HNSCC cells.” into “Mechanistically, HPV E6 might promote the expression of DNMT1 to induce PRKCZ hypermethylation, and PRKCZ increased Cdc42 levels to promote EMT in HPV+ HNSCC cells.” (See Page 20 Para 2 Line 5)
Figure Legend: We have changed the title of Figure 4 into “DNMT1 was necessary in HPV-E6 regulating PRKCZ hypermethylation”. (See Page 14 Title of Figure 4)
